# Effect of Tube Diameters and Functional Groups on Adsorption and Suspension Behaviors of Carbon Nanotubes in Presence of Humic Acid

**DOI:** 10.3390/nano12091592

**Published:** 2022-05-07

**Authors:** Mengyuan Fang, Tianhui Zhao, Xiaoli Zhao, Zhi Tang, Shasha Liu, Junyu Wang, Lin Niu, Fengchang Wu

**Affiliations:** 1State Key Laboratory of Environmental Criteria and Risk Assessment, Chinese Research Academy of Environmental Sciences, Beijing 100012, China; fangmengyuanfmy@163.com (M.F.); zth2512@163.com (T.Z.); zhaoxiaoli_zxl@126.com (X.Z.); jy.wong@foxmail.com (J.W.); 18233271321@163.com (L.N.); wufengchang@mail.gyig.ac.cn (F.W.); 2College of Geoexploration Science and Technology, Jilin University, Changchun 130026, China; 3School of Energy and Environmental Engineering, University of Science and Technology Beijing, Beijing 100083, China; liushashajida@163.com

**Keywords:** carbon nanotubes, tube diameter, functional groups, adsorption, suspension/sedimentation, humic acid

## Abstract

The adsorption and suspension behaviors of carbon nanotubes (CNTs) in the water environment determine the geochemical cycle and ecological risk of CNTs and the compounds attached to them. In this study, CNTs were selected as the research object, and the effect of tube diameters and functional groups (multiwall CNTs (MWNTs) and hydroxylated MWNTs (HMWNTs)) on the adsorption and suspension behaviors of the CNTs in the presence of humic acid (HA) was systematically analyzed. The results indicate that HA adsorption decreased with the increase in the solution pH, and the adsorption amount and rate were negatively correlated with the tube diameter of the CNTs. The surface hydroxylation of the CNTs prevented the adsorption of HA, and the maximum adsorption amounts on the MWNTs and HMWNTs were 195.95 and 74.74 mg g^−1^, respectively. HA had an important effect on the suspension of the CNTs, especially for the surface hydroxylation, and the suspension of the CNTs increased with the increase in the tube diameter. The characteristics of the CNTs prior to and after adsorbing HA were characterized by transmission electron microscopy, fluorescence spectroscopy, Fourier-transform infrared spectroscopy and Raman spectroscopy. The results indicate that surface hydroxylation of the CNTs increased the adsorption of aromatic compounds, and that the CNTs with a smaller diameter and a larger specific surface area had a disordered carbon accumulation microstructure and many defects, where the adsorption of part of the HA would cover the defects on the CNTs’ surface. Density functional theory (DFT) calculations demonstrated that HA was more easily adsorbed on the CNTs without surface hydroxylation. This investigation is helpful in providing a theoretical basis for the scientific management of the production and application of CNTs, and the scientific assessment of their geochemical cycle and ecological risk.

## 1. Introduction

The invention and development of engineered nanomaterials have been aimed to fulfill their wide applications in product manufacturing and in providing a high quality of daily life. Significant attention has been paid to engineered nanomaterials due to their diverse structures and functions and complex environmental behavior, which may impose a significant impact on the ecological environment. Approximately one million tons of engineered nanomaterials is released into the ecosystem each year, while carbon nanotubes (CNTs) are one of the ten most used engineered materials [1]. CNTs are widely used in the fields of electronics, biomaterials, medicine, cosmetics, catalysis and environmental treatment due to their unique structures, excellent electrical conductivity and superior thermal and chemical stability [2,3,4,5], and they can be used as substitutes for the scarce metals for most technologies [6]. Studies have shown that CNTs are toxic to animals, microorganisms and embryonic stem cells [7,8]. CNTs may enter and accumulate in the human body through the respiratory tract, food chain and skin contact pathways and eventually pose threats to human health [7,8,9]. Therefore, it is necessary to reduce the release of CNTs into the environment, but the high aspect ratio, aromatic structure and size of CNTs make the degradation of CNTs very difficult [10]. Hence, more attention has to be paid to CNTs for their potential risk to human health and the ecological environment.

Once released into the aquatic environment, the environmental behaviors of CNTs such as adsorption, aggregation and sedimentation could directly affect their migration, transformation, fate and bioavailability [11,12,13], and they can be significantly influenced by the physical and chemical factors of the aquatic environment, humic acid (HA) and other compounds [11,12,13]. Previous studies demonstrated that the lower pH and higher ionic strength of the water environment were favorable for the adsorption of HA on CNTs’ surfaces, but they were unfavorable for the CNTs’ stabilization [14,15,16,17]. The adsorption of HA on the surface of CNTs and CNTs’ suspension/sedimentation behaviors have been intensively studied [14,15,16]. HA plays a key role in the suspension and sedimentation of particles and colloids in the aquatic environment [18,19]. It can promote the dispersion, suspension and sedimentation of hydrophobic CNTs in the water via electrostatic repulsion, steric hindrance or solvation [11,16,18]. HA is a mixture of polyelectrolytes with different molecular weights. It contains a variety of functional groups, including carboxyl, hydroxyl, carbonyl, quinone and methoxy groups, resulting in both strong hydrophobicity and hydrophilicity [20]. HA can be adsorbed on the surface of CNTs through hydrogen bonding and hydrophobic, π−π and electrostatic interactions [11,16,20,21]. The adsorption rate is affected by the oxidation degree, specific surface area and hydrophobic force of CNTs. Meanwhile, the intraparticle diffusion is also related to the adsorption but not as a sole rate-controlling step [22].

The adsorption and suspension/settlement behaviors of CNTs in the water environment are not only influenced by the physical and chemical elements of the water, but also their own physical and chemical properties. However, limited research has been reported about the effects of the tube diameters and functional groups of CNTs on the adsorption of HA and aggregation/sedimentation of CNTs. Previous studies have demonstrated that the saturated adsorption capacity of CNTs with different structures was different [11,13,23], and hydrophilic functional groups such as -COOH or -OH on the surface of CNTs could enhance the hydrophilicity and weaken the adsorption of hydrophobic organic molecules on CNTs, thus greatly increasing the dispersion of CNTs in the water. [15,16,24]. Meanwhile, the tube diameter of CNTs may directly influence the adsorption sites, the number of functional groups and the adsorption capacity of HA. Therefore, the type and number of functional groups on the CNTs’ surface as well as the CNTs’ tube diameter will affect both the adsorption of HA and the suspension/sedimentation behavior of CNTs, which ultimately influences their migration, transformation and fate in the aquatic environment [17,25,26], affecting their environmental geochemical fate.

In this study, multiwall carbon nanotubes (CNTs) were selected as the research object, the tube diameters and functional groups were used as internal factors and the solution pH, humic acid and other physical and chemical parameters were used as external factors. The adsorption, suspension and sedimentation behaviors of CNTs were systematically studied through adsorption isotherm and kinetic experiments. Density functional theory (DFT), fluorescence excitation–emission matrix (EEM) spectroscopy, Fourier-transform infrared (FTIR) spectroscopy and Raman spectroscopy were used to study the mechanisms of the adsorption, aggregation and sedimentation of CNTs. This investigation provides the theoretical basis for recognizing the environmental behaviors and potential ecological risks of CNTs with different diameters and functional groups in the natural aquatic environment.

## 2. Materials and Methodology

### 2.1. Materials and Reagents

All reagents were analytical grade. NaOH and HCl were purchased from Sinopharm Chemical Reagent Co., Ltd., (Shanghai, China). Multiwall CNTs (MWNTs) and hydroxylated MWNTs (HMWNTs) were purchased from Chengdu Organic Chemistry Co., Ltd., Chinese Academy of Sciences. MWNTs and HMWNTs with outer tube diameters of 4–6 nm (named MWNT-1 and HMWNT-1), 5–15 nm (named MWNT-2 and HMWNT-2) and 20–30 nm (named MWNT-3 and HMWNT-3) were used to study the impact of diameters on their environmental behaviors. Humic acid (HA) was purchased from the International Humic Substances Society (IHSS). The chemical compositions, functional groups and results of the NMR analysis of HA (Elliott Soil, 1S102H) are presented in Table 1 [27]. Deionized water (DI water) was supplied by an ultra-pure water system (Milli-Q Advantage System, Millipore, Boston, MA, USA). The water resistance value was ≥18.3 MΩ·cm, and the conductivity was ≤10 us·cm^−1^.

### 2.2. Preparation of HA Stock Solution and Determination of Standard Curve for HA Concentration

HA was dissolved in 0.1 M NaOH solution. Then, the pH of the solution was adjusted to 7.9 ± 0.2 with 0.1 M HCl. The solution was kept in a shaker at room temperature (22 ± 1 °C) for 24 h to dissolve completely and filtered through a 0.45 μm fiber membrane (MF Cat No: HAWP04700) prior to use. HA solutions with concentrations between 5.0 and 80.0 mg L^−1^ were prepared by diluting the HA stock solution. The concentration of the HA solution was measured using a UV–visible spectrophotometer (UV-vis 8453, Agilent, Palo Alto, CA, USA)with a 1 cm-light-path quartz cuvette. The light absorbance at a fixed wavelength of 254 nm was used for establishing the HA calibration curve.

### 2.3. Characterization of CNTs

Transmission electron microscopy (TEM; H-7500 Hitachi, Tokyo, Japan) was used to observe the morphology, tube diameter and length of CNTs and CNTs-HA. TEM images were recorded on an H7500 transmission electron micrograph (Hitachi, Tokyo, Japan) operated at 120 kV. All CNT samples used for TEM testing were prepared by cool drying of the solution containing suspended CNTs. Zeta potentials of CNTs were measured at various pH values using a Nano-ZS90 Zetasizer (Malvern Instruments, Worcestershire, UK).

### 2.4. Effect of Solution pH

An amount of 4.0 mg of CNTs was mixed with 20 mL of DI water in a 100 mL polycarbonate bottle aided by sonication for 30 min (240 W, 100 kHz). The temperature was maintained at 22 ± 1 °C using cooling water. Then, 20.0 mL of HA solution (40.0 mg L^−1^) was added, while the solution pH was adjusted in the range of 2.0 to 10.0 using 0.1 M HCl or NaOH. The mixture was stirred on a rotary shaker for 48 h. Then, the supernatant was filtered through a 0.2 μm fiber membrane (PALL REF 4612) for the absorbance measurement using a UV–Vis spectrophotometer (UV-vis 8453, Agilent, Palo Alto, CA, USA) (manufacturer, city, (State or Province), country) at 254 nm. The experiment was repeated three times.

### 2.5. Adsorption Isotherm

The sonicated CNT suspension solution was prepared as described in Section 2.4. Then, 20.0 mL of HA solution (200.0 mg L^−1^) was added to achieve final HA concentrations of 5.0, 10.0, 15.0, 20.0, 30.0, 40.0 and 50.0 mg L^−1^. The solution pH was adjusted to 6.0 using 0.1 M HCl. The mixture was stirred in a rotary shaker for 48 h. Then, the supernatant was filtered through a 0.2 μm fiber membrane (PALL REF 4612) for the absorbance measurement using a UV–Vis spectrophotometer at 254 nm. The adsorption isotherms were fitted using Langmuir and Freundlich adsorption isotherm models. The experiment was repeated three times.

### 2.6. Adsorption Kinetics

An identical mixture solution was prepared as described in Section 2.4, and the initial concentration of HA was 20.0 mg L^−1^. The solution pH was adjusted to 6.0 using 0.1 M HCl. The solution was then shaken for 5 min, 30 min, 60 min, 2 h, 4 h, 6 h, 12 h, 24 h, 48 h, 72 h, 96 h and 120 h. Then, the supernatant was filtered through a 0.2 μm fiber membrane (PALL REF 4612) for the absorbance measurement using a UV–Vis spectrophotometer at 254 nm. The experiment was repeated three times.

### 2.7. Sedimentation Test of CNTs

The sonicated CNT suspension solution was prepared as described in Section 2.4. Then, 20 mL of HA solution (200.0 mg L^−1^) was added to achieve final HA concentrations of 5.0, 10.0, 20.0, 30.0, 40.0, 50.0 and 60.0 mg L^−1^. The solution pH was adjusted to 6.0 using 0.1 M HCl. After shaking for 48 h, the sedimentation of the suspensions was measured with a UV–visible spectrophotometer at 800 nm [14,25]. The ratio of the absorbance at different times (C_e_) to the initial absorbance (C_0_) as a function of time was used to determine the sedimentation dynamics. A lower C_e_/C_0_ ratio indicates a higher sedimentation performance of the nanotubes and, in turn, much easier aggregation and settlement of the CNTs [28].

### 2.8. Fluorescence Spectral Analysis

The initial concentration of HA was 20.0 mg L^−1^, and the fluorescence spectra of the remaining HA after being adsorbed by CNTs at 24 h, 48 h and 72 h were measured and compared with the fluorescence spectra of the initial HA. Fluorescence spectra of HA were measured using a fluorescence spectrometer (Hitachi F-7000, Tokyo, Japan) with a 1 cm-path-length quartz cuvette at room temperature. EEM spectra were obtained by subsequently scanning emission (Em) wavelengths from 230 to 600 nm and excitation (Ex) wavelengths from 200 to 450 nm, both stepped by 5 nm intervals. Slit widths were 5 nm for both Ex and Em, and the scanning speed was set at 12000 nm∙min^−1^. The fluorescence index (FI) was calculated as the ratio of the emission intensity at Em 450 nm relative to that at Em 500 nm at Ex 370 nm [29].

### 2.9. Raman and FTIR Spectroscopy Characterization of CNTs

A micro-Raman imaging spectrometer (DXRxi, Thermo Scientific, Waltham, MA, USA) was used to characterize the CNTs and CNTs-HA. The microscope was equipped with 10× and 50× objectives. During the measurement, a layer of CNTs with a thickness of 2 cm was placed on a glass slide mounted horizontally inside the test chamber. The orientation of the sample stage was adjusted so that the laser spot scanned in parallel. The excitation source had a wavelength of 532 nm. The focused spot size was about 1 μm. The spectral resolution was 1 μm. The optical path length was corrected by the Raman peak intensity at 520 cm^−1^ from a silicon wafer with an excitation power of 2 mW and an exposure time of 10 s. The test results were analyzed and processed with Omnic software with baseline correction and Lorentz peak fitting for peak deconvolution. FTIR spectrometry of CNTs and CNTs-HA was carried out using a Nicolet Magna-IR 750 FTIR spectrometer (Nicolet Magna-IR 750, Nicolet, Madison, WI, USA) using KBr powder as the background. FTIR spectra were recorded from 400 to 4000 cm^−1^ at a resolution of 4 cm^−1^ and averaged over 200 scans.

### 2.10. Atomic Adsorption Theory Analysis Based on Density Functional Theory (DFT) Calculation

Density function theory calculations were performed using the CP2K package [30]. The PBE functional [31] with Grimme D3 correction [32] was used to describe the system. Unrestricted Kohn–Sham DFT was used as the electronic structure method in the framework of the Gaussian and plane wave methods [33,34]. The Goedecker–Teter–Hutter (GTH) pseudopotentials [35,36] and DZVPMOLOPT-GTH basis sets [33] were utilized to describe the molecules. A plane-wave energy cut-off of 500 Ry was employed.

## 3. Results and Analysis

### 3.1. Characterization of CNTs

The TEM images of the MWNTs (Figure 1a–c) and HMWNTs (Figure 1d–f) prior to and after adsorbing HA (Figure 1g–l) are shown in Figure 1. The observed outer tube diameters of the CNTs were consistent with the data given by the manufacturer. Additionally, it can be seen that the MWNTs and HMWNTs exhibited the characteristic of flexible winding, and the winding property of the MWNTs and HMWNTs increased significantly after the adsorption of HA. This result is different from that of nanoparticles, which have better dispersion after HA adsorption [14,26]. Previous research indicated that the tube segments of CNTs are usually closed, and the inner diameter of MWNTs and HMWNTs is too small to allow large molecules such as HA to enter; hence, HA can only be adsorbed on the outer surface of MWNTs and HMWNTs [16,17].

The specific surface areas (BET) after HA adsorption of MWNT-1, MWNT-2, MWNT-3, HMWNT-1, HMWNT-2 and HMWNT-3 were 342.4, 213.8, 129.5, 471.6, 220.4 and 135.1 m^2^ g^−1^, respectively (Appendix A). For the CNTs with different surface functional groups, their BET also increased with the decrease in the tube diameter. After the adsorption of HA, the BET of MWNTs and HMWNTs decreased significantly; the reduction rates of MWNTs (19.9–30.9%) were higher than those of HMWNTs (12.0–25.7%), indicating that the adsorption capacity of MWNTs for HA was larger than that of HMWNTs; and HA occupied more adsorption sites of MWNTs. The nitrogen adsorption isotherms for the MWNTs and HMWNTs are shown in Appendix A.

The effect of the solution pH on the changes in the surface charge property and density of the MWNTs and HMWNTs are shown in Figure 2. The MWNTs and HMWNTs had a negative surface charge under different pH conditions, and only MWNT-1 and MWNT-2 had small positive charges at pH < 3.0. With the increase in pH, more OH^−^ adsorbed on the surface of MWNT-1 and MWNT-2, and the positive charge density decreased, resulting in zero surface charge at pH 3.0. The surface negative charge density of MWNT-1 and MWNT-2 increased gradually with the increase in the solution pH. The zeta potentials of the MWNT-3 and HMWNT surfaces were negative. With the increase in pH, the negative value of the zeta potential increased. The results indicate that the surface functional groups and tube diameters of the CNTs also affected the property and density of the surface charge. For different surface functional groups, the surface negative charge density of the MWNTs and HMWNTs also increased with the increase in the solution pH and tube diameter.

### 3.2. Effect of pH on the Adsorption of HA

The adsorption amount of HA on the surface of the MWNTs and HMWNTs decreased significantly with the increase in the solution pH (Figure 3); especially for the MWNTs and HMWNTs with a larger diameter, HA could hardly be adsorbed on their surface when the solution pH was greater than 7.0, and the maximum adsorption amount of HA was achieved at the lowest pH of 2.0. The adsorption amount of HA on the surface of the MWNTs was slightly higher than that of the HMWNTs, consistent with the above results of BET. For the MWNTs, the maximum adsorption amount of HA decreased with the increase in the tube diameter following the order of MWNT-1 (186.4 mg g^−1^) > MWNT-2 (181.7 mg g^−1^) > MWNT-3 (127.4 mg g^−1^). Similar results were also found for the HMWNTs, with the adsorption amount decreasing in the order of HMWNT-1 (181.1 mg g^−1^) ≈ HMWNT-2 (182.6 mg g^−1^) > HMWNT-3 (120.8 mg g^−1^). Therefore, the adsorption amount of HA was greater with smaller tube diameters regardless of the functional groups on the surface of the CNTs. Previous studies demonstrated that the solution pH not only affected the property and density of the surface charge of CNTs, but also the dissociation of HA in the solution [14,16]. Therefore, interactions between the surface functional groups of CNTs and the hydrogen bonds and polar functional groups of HA were inhibited, thus reducing the adsorption amount of HA [14]. The results indicate that the surface functional groups had no significant effect on the adsorption of HA on the CNTs at the low solution pH, and the surface hydroxylation hindered the adsorption of HA at the high solution pH.

After the adsorption of HA, the zeta potentials of the MWNTs and HMWNTs decreased first and then increased (Figure 4). This phenomenon can be explained from two aspects: on the one hand, HA was adsorbed on the surface of the CNTs at a low solution pH and then increased the negative charge density of the CNTs due to its negative charge in the solution; on the other hand, under the condition of a high solution pH, HA was difficult to adsorb on the surfaces of the MWNTs and HMWNTs, and the zeta potentials of the MWNTs and HMWNTs were only affected by the solution pH.

### 3.3. Adsorption Isotherms

The adsorption of HA on the MWNTs and HMWNTs was fitted using both Langmuir (Equation (1)) and Freundlich (Equation (2)) [37,38] adsorption models (Figure 5).
(1)$$q_{e}=\frac{q_{m}k_{L}C_{e}}{1+k_{L}C_{e}}$$
(2)$$q_{e}=k_{F}C_{e}{}^{\frac{1}{n}}$$
where q_e_ is the amount of HA adsorbed at equilibrium (mg g^−1^); C_e_ is the concentration of HA in the solution (mg L^−1^) at equilibrium; q_m_ (mg g^−1^) is the maximum adsorption capacity; k_L_ (L g^−1^) is the Langmuir equilibrium constant; k_F_ (mg^1−(1/n)^L^1/n^ g^−1^) and n are the Freundlich parameters.

The Langmuir and Freundlich adsorption isotherms of HA on the MWNT and HMWNT surfaces were obtained by fitting Equations (1) and (2) (Table 2). In general, most of the isotherms achieved a good-quality fitting. Based on the Langmuir isotherm, the maximum adsorption amount of HA on the surface of MWNT-1 was 195.95 mg g^−1^ at pH 6.0. However, under the same conditions, this was reduced to 89.73 and 87.99 mg g^−1^ for MWNT-2 and MWNT-3, respectively. Therefore, the MWNT with the smaller tube diameter and the larger specific surface area had the highest adsorption capacity for HA. The maximum adsorption amounts of HA on the surfaces of HMWNT-1, HMWNT-2 and HMWNT-3 were 74.74, 75.16 and 69.81 mg g^−1^, respectively. The surface hydroxylation of the HMWNTs greatly reduced the adsorption amount of HA, and the effect of the tube diameter on the adsorption of HA on the HMWNTs was not as significant as that on the MWNTs. The adsorption amount of HA on the MWNTs was larger than that on the HMWNTs, which could be attributed to the fact that the hydroxyl groups in the HMWNTs occupied a part of the adsorption sites, so the surface hydroxylation of the MWNTs may inhibit the adsorption of HA. The constant “*n*” for the strength of adsorption predicted using the Freundlich equation was less than 2.0, indicating that the MWNTs and HMWNTs had a weak adsorption capacity for HA. This is understandable, since at pH 6.0, the surfaces of the MWNTs and HMWNTs were negatively charged, and a part of the HA can be adsorbed on the surfaces of the MWNTs and HMWNTs due to electrostatic repulsion.

To better understand the adsorption behavior of HA on the surfaces of the MWNTs and HMWNTs, Temkin (Equation (S1)) (Appendix A) and Dubinin–Radushkevich (Equations (S2) and (S3)) [37,38] (Appendix A) adsorption isotherms were used to analyze the adsorption of HA. The results of the Temkin and Dubinin–Radushkevich adsorption isotherms are listed in Appendix A. Correlation coefficient calculations showed that the Temkin isotherm represents the equilibrium data of 298 K well. The results indicate that the binding energy of HA was evenly distributed at 298 K. Compared with the data in Table 2, the order of the fitting results of each model was R^2^_L_> R^2^_F_> R^2^_T_> R^2^_D-R_. This indicates that the adsorption of HA on the MWNTs and HMWNTs was mainly monolayer adsorption [39].

### 3.4. Adsorption Kinetics

The tube diameter may influence the adsorption pathway for CNTs, where a smaller tube diameter of CNTs offers a shorter adsorption pathway. Therefore, the study of the adsorption kinetics is helpful for a better understanding of the adsorption of HA on the surfaces of CNTs with different tube diameters and functional groups. As shown in Figure 6, the adsorption amount of HA on the MWNTs was higher than that on the HMWNTs at pH = 6.0, and the adsorption equilibrium times of HA on the surfaces of the MWNTs and HMWNTs were consistent. The adsorption amount of HA increased rapidly in the first 24 h, which slowed down gradually after 24 h until reaching adsorption equilibrium at about 48 h. For the same surface functional groups of the CNTs, the adsorption amount of HA decreased with the increase in the tube diameter, consistent with the result of the adsorption isotherm fitting. Therefore, the CNTs with a larger specific surface area and a shorter adsorption path had a greater adsorption rate of HA.

In order to quantify the time-dependent variation in the HA adsorption on the MWNT and HMWNT surfaces, pseudo-first-order (Equation (3)) (Appendix A) and pseudo-second-order (Equation (4)) (Figure 7) kinetic models were used for fitting the time-dependent profiles [37,38]. The adsorption kinetic parameters of HA can be calculated by measuring the adsorption rate of HA on MWNTs and HMWNTs.

Pseudo-first-order model:(3)$$q_{t}=q_{e}(1-e^{-k_{1}t})$$

Pseudo-second-order model:(4)$$\frac{t}{q_{t}}=\frac{1}{k_{2}q_{e}{}^{2}}+\frac{1}{q_{e}}t$$
where k_1_ (min^−1^) and k_2_ (g mg^−1^ min^−1^) are the first-order and second-order adsorption rate constants, q_t_ (mg g^−1^) is the amount of HA adsorbed by the CNTs at time t and q_e_ (mg g^−1^) is the adsorption capacity at adsorption equilibrium. The initial adsorption rate h_0_ (mg/g/min) can be defined as follows (Equation (5)):(5)$$h_{0}=k_{2}q_{e}{}^{2}(t\to 0)$$

Both k_2_ and h_0_ can be determined experimentally by plotting t/q_t_ against t.

The intraparticle diffusion rate was obtained from the plots of q_t_ versus t^1/2^. The rate parameter for intraparticle diffusion was determined using the following Equation (6) [40]:(6)qt=kintt12+C
where C is the intercept, and k_int_ is the intraparticle diffusion rate constant (mg g^−1^ min^−1/2^).

From Appendix A, the regression of q_t_ versus t^1/2^ was linear, but none of the straight lines passed through the origin, indicating that the intraparticle diffusion was not the only rate-controlling step.

The adsorption kinetics of HA on the surface of the MWNTs and HMWNTs were in accordance with the second-order kinetic model (R^2^ = 0.996–0.999) (Table 3), which indicates that chemical adsorption was the rate-limiting step for HA absorbed on the surface of the MWNTs and HMWNTs. The adsorption kinetic analysis results also demonstrate that the adsorption rates of HA on the MWNT-1 and MWNT-2 surfaces were similar, but much higher than that of MWNT-3. The adsorption rate constant and maximum adsorption amount of HA on HMWNT-1 were also much higher than those of other HMWNTs with a larger tube diameter. By comparing the adsorption rates of HA on the MWNTs with different surface functional groups, it can be seen that surface hydroxylation reduced not only the adsorption amount of HA, but also the adsorption rate. Therefore, the surface functional groups and tube diameters of the CNTs also played a significant role in the adsorption of HA, and the specific surface area, tube diameter and interaction between the surface functional groups and HA synergistically affected the adsorption rate and adsorption capacity of HA.

The values of C and k_int_ are presented in Table 3. The values of k_int_ for the MWNTs were generally higher than those for the HMWNTs, revealing that the adsorption rate of HA on MWNTs was higher. This result is consistent with the conclusion of the second-order kinetic model. The values of C were proportional to the extent of the boundary layer thickness, that is, the larger the intercept, the greater the boundary layer effect [40]. The values of C decreased with the increasing tube diameters of the MWNTs and HMWNTs, indicating that the CNTs with a smaller outer tube diameter had a larger initial adsorption amount. The C values of MWNT-1 (11.907 mg g^−1^) and HMWNT-1 (12.478 mg g^−1^), and MWNT-2 (8.476 mg g^−1^) and HMWNT-2 (8.289 mg g^−1^), were close. The results suggest that hydroxylation had little effect on the initial adsorption amount of HA on the CNTs with a smaller outer tube diameter [41]. For the MWNTs with larger outer tube diameters, hydroxylation had a greater effect on the initial adsorption.

### 3.5. Effect of HA on CNT Suspension/Sedimentation

In the water environment, the suspension and sedimentation behaviors of nanomaterials greatly affect their migration, transformation, fate and ecological effects. Therefore, the study of the suspension and sedimentation behaviors of MWNTs and HMWNTs with different tube diameters and surface functional groups is of great significance for understanding the ecological risks of CNTs. The effects of the HA concentration on the suspension/sedimentation behaviors of the MWNTs and HMWNTs are shown in Figure 8. In the absence of HA, the suspension of the MWNTs was less stable than that of the HMWNTs with the same tube diameter, which was mainly due to the higher surface negative charge density. The effect of the tube diameter on the suspension stability of the MWNTs and HMWNTs was similar, and the CNTs with a smaller tube diameter were more likely to aggregate and settle. The aggregation of the MWNTs and HMWNTs increased their effective hydraulic diameter with reduced surface tension and intensified sedimentation. Thus, the CNTs with a smaller tube diameter were more likely to aggregate and sediment in the same water environment.

In the presence of HA, the suspension of the CNTs significantly improved with the increase in the HA concentration. Under the condition of pH = 6.0, the surface of the CNTs was negatively charged, and HA was adsorbed on the surface of the CNTs through hydrophobic and π–π interactions [24,26]. The adsorption of HA had two effects on the suspension of the CNTs. On the one hand, HA increased the surface negative charge density of the CNTs, which increased the electrostatic repulsion between the CNTs. On the other hand, the adsorption of HA increased the steric hindrance between the CNTs. The concentration of HA greatly affected the suspension behavior of the CNTs, and the suspension performance of the CNTs with different surface functional groups and tube diameters increased with the increase in the HA concentration. For different surface functional groups, the adsorption of HA had a greater effect on the suspension of the CNTs due to the higher HA adsorption amount on the CNTs. For different tube diameters, the effect of HA on the suspension performance of the CNTs with a smaller tube diameter was higher than that of the CNTs with a larger tube diameter. Therefore, HA significantly promoted the suspension of the MWNTs and HMWNTs, and the adsorption amount of HA was an important factor. CNTs are more easily suspended in the aquatic environment with a higher concentration of natural organic matter, which leads to a greater impact on the carbon cycle and a higher ecological risk.

### 3.6. Fluorescence Spectral Analysis

The fluorescence spectra of HA prior to and after being adsorbed on the surfaces of the MWNTs and HMWNTs are shown in Appendix A. The characteristic peak of each spectrum at Ex/Em = 275 nm/510 nm (abbreviated as peak A) is very obvious and has a wide spectrum coverage. Peak A of HA significantly reduced after the adsorption, which indicates that HA was successfully adsorbed on the surface of the MWNTs and HMWNTs. The low fluorescence index (FI) values are rich in aromatic moieties [29,42,43]. The FIs of the residual HA after being adsorbed on the CNTs with different surface functional groups and tube diameters were compared (Figure 9). With the increase in the adsorption time, the FI of the residual HA decreased, indicating that the aromatic groups of HA were more difficult adsorb on the surface of the CNTs than other parts, especially for the MWNTs. Therefore, the surface hydroxylation of the CNTs may have increased the adsorption of aromatic compounds of HA. Meanwhile, the FI of the residual HA increased with the increase in the tube diameter of the CNTs, thus the MWNTs and HMWNTs with larger tube diameters were more likely to adsorb the aromatic groups of HA. Therefore, the tube diameters and surface functional groups of CNTs not only affect the adsorption amount of HA, but also the components of HA.

### 3.7. Raman Spectroscopy and FTIR Study

The Raman spectra of the MWNTs and HMWNTs prior to and after adsorbing HA are shown in Figure 10. After HA was adsorbed on the surfaces of the MWNTs and HMWNTs, there were no significant changes in the positions of the CNTs’ characteristic peaks. The peaks at 1570 cm^−1^ (G peak) and 1340 cm^−1^ (D peak) represent ordered and disordered carbon stacking microstructures within the CNTs, respectively. Previous studies indicated that HA also contributed, with Raman peaks at 1379 and 1590 cm^−1^ originating from the symmetric vibration of the carboxyl groups, C=C bond vibration of the carboxyl group in the aromatic group, C-O vibration of the phenolic group and phenol vibration of HA [44]. As the characteristic peaks of the CNTs and HA overlapped to some degree, no significant changes were found in the Raman spectra of the MWNTs and HMWNTs prior to and after HA adsorption. However, it can be seen that the intensities of the characteristic peak significantly increased after HA was adsorbed on the surfaces of the MWNTs and HMWNTs, which indirectly proves the adsorption of HA.

The ratio of the D-band to G-band intensities (I_D_/I_G_) can be used to characterize the degree of defects of CNTs. A higher I_D_/I_G_ value indicates more defects on CNTs [45,46]. The I_D_/I_G_ values of MWNT-1 and HMWNT-1 were 1.17 and 1.10 (Table 4), respectively, indicating that the MWNTs had a more disordered structure, and the corresponding specific surface area was also larger. After the adsorption of HA, the Raman peak intensities of MWNT-1 and HMWNT-1 had the largest increase, accompanied by the increase in I_D_/I_G_. After the adsorption of HA, the intensity of the characteristic peak in the Raman spectrum increased the most, and the value of I_D_/I_G_ also increased, which indicates that the CNTs with the smallest diameter had the largest adsorption capacity for HA. The reason for this result may be the coincidence of the characteristic peaks of the CNTs and HA. The I_D_/I_G_ values of MWNT-2 and MWNT-3 were smaller than that of MWNT-1, indicating fewer defects and a smaller specific surface area. The I_D_/I_G_ values of HMWNT-2 and HMWNT-3 were very different from that of HMWNT-1, in agreement with the above results of the adsorption isotherms. After HA was adsorbed on the HMWNTs, the value of I_D_/I_G_ decreased due to HA occupying the defect sites. In addition to the G and D peaks, there were two weak Raman peaks, namely, the 2D peak at 2680 cm^−1^ and the D + G peak at 2940 cm^−1^. The 2D and D + G peaks represent defect-free and disordered carbon accumulation microstructures, respectively. After HA was adsorbed on the MWNTs and HMWNTs, the I_2D_/I_D+G_ values decreased, indicating that the number of defects increased after the adsorption of HA. This observation is in contrast to the change in the I_D_/I_G_ value prior to and after the adsorption of HA. As the peak heights of the 2D and D + G peaks were much smaller than those of the D and G peaks, the I_D_/I_G_ value should prevail.

In order to explain the mechanism of adsorption, Fourier-transform infrared spectroscopy (FTIR) was used to study the surface chemical reactions during adsorption (Figure 11). The spectrograms of each CNT had two obvious characteristic peaks near 1160 and 3430 cm^−1^, and 3430 cm^−1^ corresponds to the stretching vibration absorption peak of the -OH bond [47]. The absorption peak near 1160 cm^−1^ is the stretching vibration peak of C-O in the carboxyl group [48], and the peak increased significantly after the adsorption of HA. In addition, each CNT had a weak peak near 2915 cm^−1^ and 1400 cm^−1^, which represent the tensile vibration of the benzene ring and C-C in the carbon nanotube [49]. The peak at 1700 cm^−1^ represents the vibration peak of C=O in the carboxyl group [49,50], indicating that the CNTs were grafted to carboxyl groups on the surface. Peaks at 1625 cm^−1^ were observed for MWNT-2 and HMWNT-2, and the peak strength decreased after HA adsorption, indicating the release of hydroxyl groups [48]. The vibration peak at 1575 cm^−1^ represents the E1u vibration mode of the carbon nanotube wall, which is caused by the stretching vibration of the C=C skeleton in the carbon ring [47], indicating the existence of a graphite structure in the CNTs. The infrared spectra of the CNTs after HA attachment showed an obvious vibration peak at 1575 cm^−1^, indicating that the overall structure of the CNTs was not damaged to a large extent after HA adsorption.

### 3.8. Atomic Adsorption Theory Analysis Based on DFT

The crystal surface model of the carbon nanotubes was based on the first principles of density functional theory (DFT). The simulation was carried out in a cubic box of 20 ×19.67 × 36.00 Angstrom^3^. There were 16 -OH groups that were used to modify the carbon nanotube CNT(8,8). Figure 12a shows the atomic arrangement of HA absorbed on the MWNTs, where C-C was neatly arranged in the MWNTs. Figure 12b shows the atomic arrangement of HA absorbed on the HMWNTs, with an O atom bonded to a H atom, which was uniformly attached to the C atom in the HMWNTs. HA molecules were uniformly adsorbed on the surface of the MWNTs and HMWNTs, and HA was parallel to the surface of the MWNTs and HMWNTs. Figure 12c shows the charge difference between the HA molecules and HMWNTs.

The dark gray, light gray and red balls represent carbon, hydrogen and oxygen atoms, respectively.

The adsorption energy (Ead) value determines the adsorption stability, and the adsorption energy of HA molecules was calculated by Equation (7).

The adsorption energy is defined as
E_a_ = E_mol/sur_ − E_mol_ − ρ_sur_(7)
where E_mol/sur_, E_mol_ and E_sur_ are the DFT energy of the molecule adsorbed on surface, and of the molecule and surface.

The charge density difference is defined as
Δρ = ρ_mol/sur_ − ρ_mol_ − ρ_sur_(8)
where ρ_mol/sur_, ρ_mol_ and ρ_sur_ are the electron density of the molecule adsorbed on the surface, and the individual electron densities of the molecule and surface.

The adsorption energy of the molecule on HMWNTs (8,8) was about −2.40 eV, while that on MWNTs (8,8) was about −1.45 eV. This demonstrates that HA was more likely to be adsorbed on the surface of the MWNTs, and that the adsorption was relatively stable, which is also consistent with the results of the adsorption isotherm studies. Therefore, in the natural aquatic environment, MWNTs are more likely to absorb organic pollutants than HMWNTs, resulting in a higher ecological risk.

## 4. Conclusions

The adsorption amount of HA on both the MWNT and HMWNT surfaces gradually decreased with the increase in the solution pH and tube diameters. At the same tube diameter, the surface hydroxylation prevented the adsorption of HA on the surface of the CNTs. The adsorption rate on the surface of the MWNTs was much higher than that on the surface of the HMWNTs. The suspension of the HMWNTs was higher than that of the MWNTs with the same tube diameter, and the suspension of the CNTs increased with the increase in the tube diameter. The adsorption amount of HA on the surface of the CNTs determined the suspension behavior of the CNTs. Compared with the MWNTs, the HMWNTs could more easily adsorb the aromatic moieties of HA. The adsorption of HA reduced the surface defects of the MWNTs and HMWNTs. Therefore, a larger diameter and surface hydroxylation enhanced the suspension of the CNTs, and HA had a positive effect on the suspension behavior of the CNTs in the water environment and enhanced the migration ability of the CNTs, leading to higher potential ecological risks.

## Figures and Tables

**Figure 1 nanomaterials-12-01592-f001:**
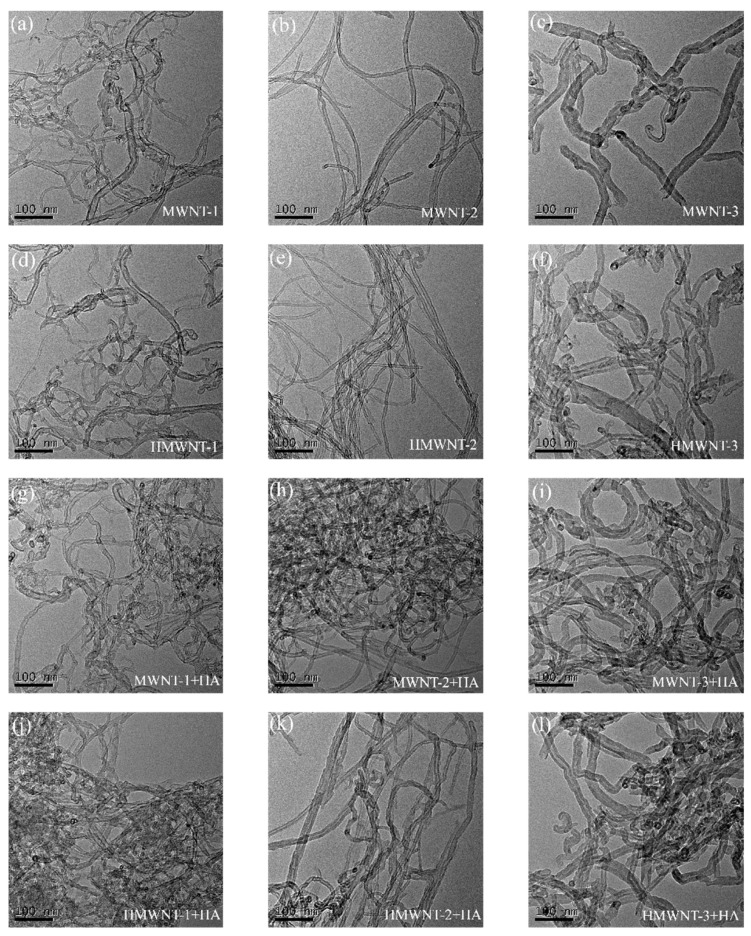
TEM images of MWNTs (**a**–**c**,**g**–**i**) and HMWNTs (**d**–**f**,**j**–**l**) prior to and after HA adsorption.

**Figure 2 nanomaterials-12-01592-f002:**
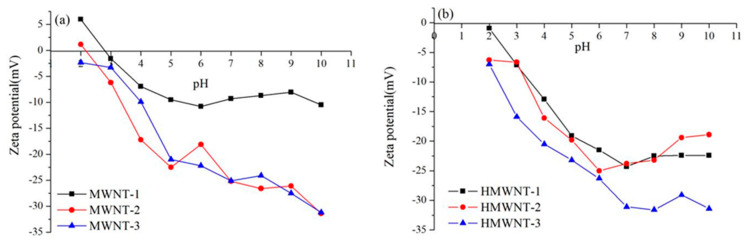
Effect of pH on the surface charge property and density of MWNTs (**a**) and HMWNTs (**b**).

**Figure 3 nanomaterials-12-01592-f003:**
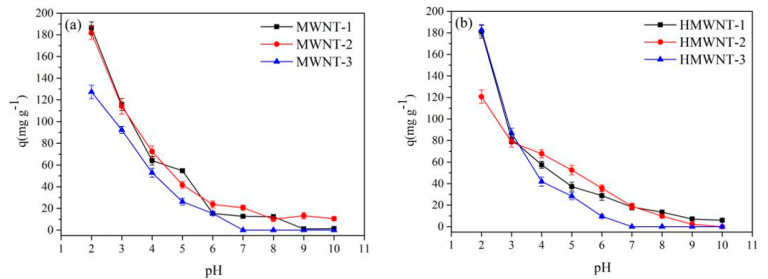
The effect of the solution pH on HA adsorption on the surfaces of MWNTs (**a**) and HMWNTs (**b**).

**Figure 4 nanomaterials-12-01592-f004:**
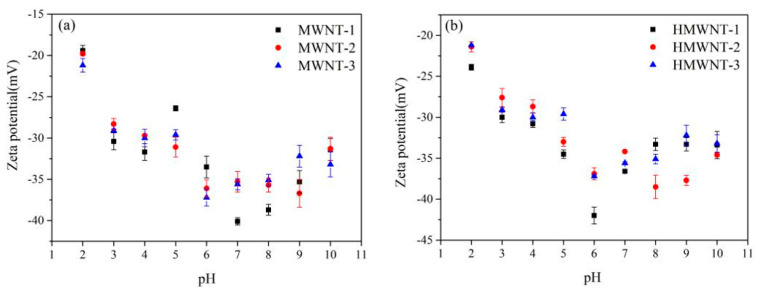
Zeta potential changes after HA adsorption of MWNTs (**a**) and HMWNTs (**b**) at different solution pH values.

**Figure 5 nanomaterials-12-01592-f005:**
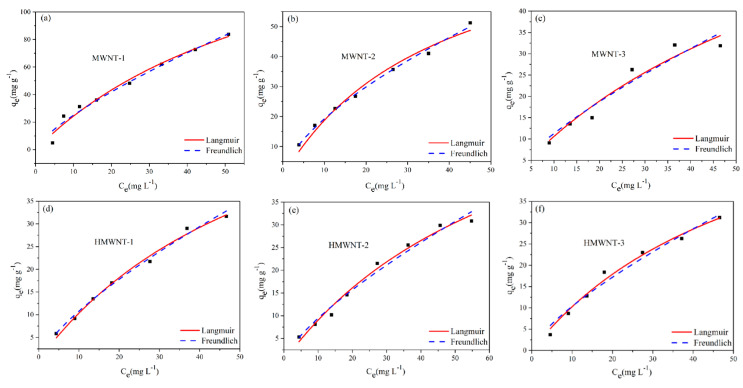
Langmuir and Freundlich adsorption isotherms for HA adsorption on MWNTs (**a**–**c**) and HMWNTs (**d**–**f**).

**Figure 6 nanomaterials-12-01592-f006:**
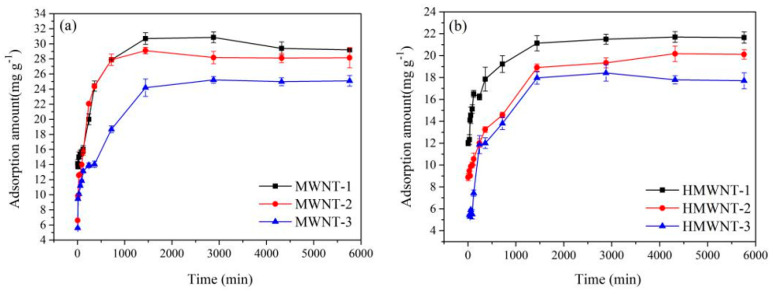
HA adsorption on MWNT (**a**) and HMWNT (**b**) surfaces under different equilibrium times.

**Figure 7 nanomaterials-12-01592-f007:**
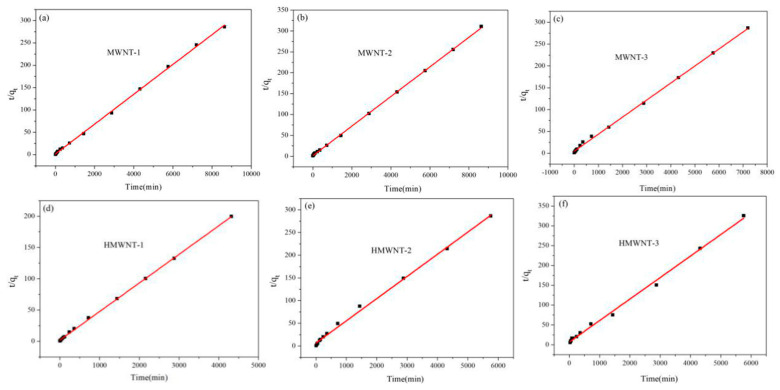
Pseudo-second-order kinetic curves of the adsorption of HA on the surface of MWNTs (**a**–**c**) and HMWNTs (**d**–**f**).

**Figure 8 nanomaterials-12-01592-f008:**
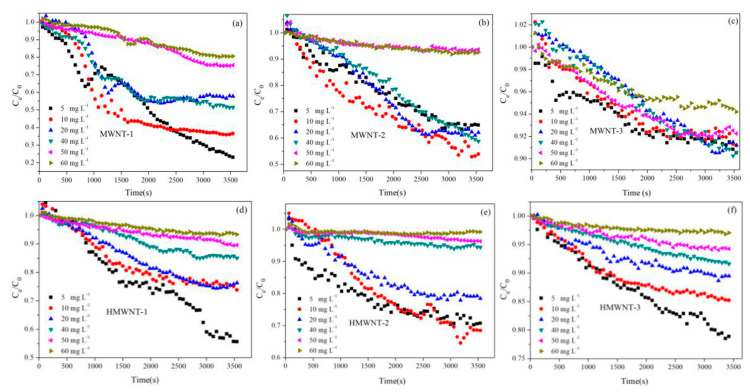
Effect of HA adsorption on sedimentation behaviors of MWNTs (**a**–**c**) and HMWNTs (**d**–**f**).

**Figure 9 nanomaterials-12-01592-f009:**
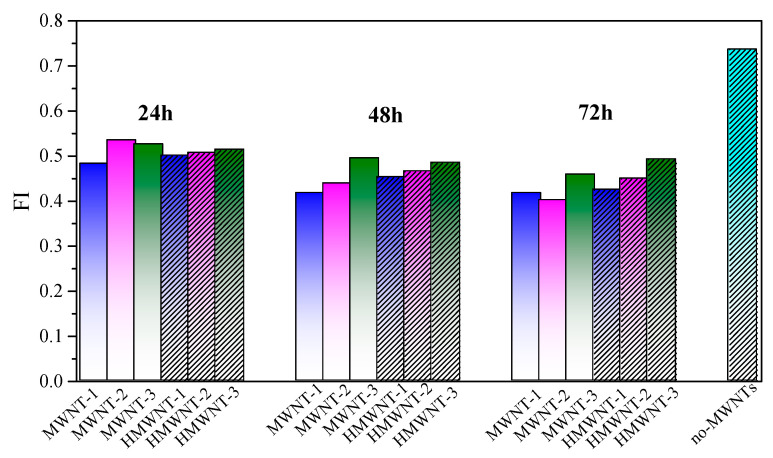
Changes in the florescence index (FI) for HA after adsorption on MWNTs and HMWNTs.

**Figure 10 nanomaterials-12-01592-f010:**
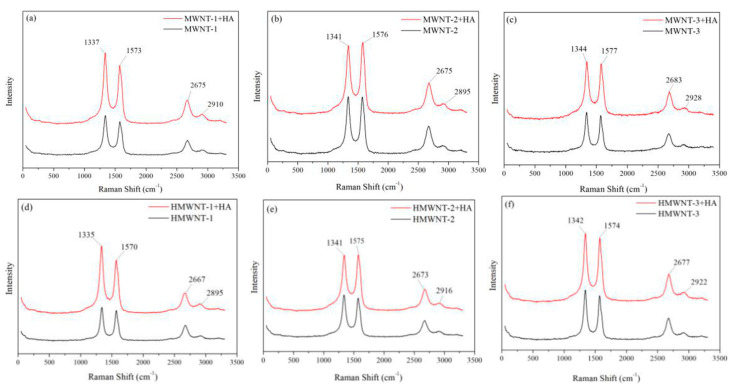
Raman spectra of MWNTs (**a**–**c**) and HMWNTs (**d**–**f**) prior to and after HA adsorption.

**Figure 11 nanomaterials-12-01592-f011:**
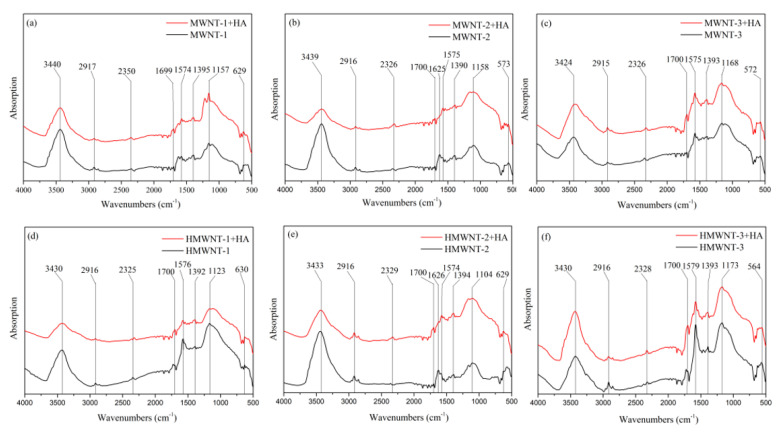
FTIR spectra of MWNTs (**a**–**c**) and HMWNTs (**d**–**f**) prior to and after HA adsorption.

**Figure 12 nanomaterials-12-01592-f012:**
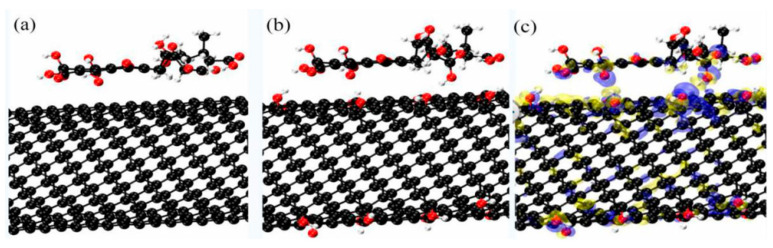
DFT for HA adsorbed on MWNT (**a**) and HMWNT facets (**b**), and the charge difference between the HA molecules and HMWNTs (**c**).

**Table 1 nanomaterials-12-01592-t001:** The elemental composition, functional groups and NMR analysis of HA.

Carbon Distribution (mg L^−1^)								
Sample	Carbonyl 220–190	Carboxyl 190–165	Aromatic 165–110	Acetal 110–90	Hetero Aliphatic 90–60		Aliphatic 60–0	Aromatic/Aliphatic
HA	6	18	50	4	6		16	3.125
**Element Constitution %** **(** **w·w^−1^** **)**								
	H_2_O	Ash	C	H	O	N	S	P
HA	8.2	0.88	58.13	3.68	34.08	4.14	0.44	0.24
**Acid Functional Groups** **(** **m mol·g^−1^** **)**								
	Carboxyl	Phenolic	Q_1_	LogK_1_	N_1_	Q_2_	LogK_2_	N_2_
HA	8.28	1.87	8.90	4.36	3.16	0.85	9.80	1.00

Notes: The data were sourced from the International Humic Substances Society (IHSS). Q_1_ and Q_2_ are the maximum charge densities of the two classes of binding sites; LogK1 and LogK2 are the mean logK values for proton binding by the two classes of sites; N is the number of fitted titration data points.

**Table 2 nanomaterials-12-01592-t002:** Langmuir and Freundlich adsorption isotherm constant of HA on MWNTs and HMWNTs.

Samples	Langmuir			Freundlich		
	q_m_ (mg g^−1^)	K_L_ (L mg^−1^)	R^2^	K_F_ (mg^1−(1/n)^L^1/n^g^−1^)	n	R^2^
MWNT-1	195.95	0.0142	0.975	4.50	1.341	0.974
MWNT-2	89.73	0.0264	0.977	4.48	1.578	0.995
MWNT-3	87.99	0.0137	0.934	2.08	1.362	0.998
HMWNT-1	74.74	0.0161	0.991	2.04	1.383	0.996
HMWNT-2	75.16	0.0136	0.985	1.74	1.362	0.977
HMWNT-3	69.81	0.0172	0.985	1.92	1.368	0.971

**Table 3 nanomaterials-12-01592-t003:** Adsorption kinetic parameters of HA on MWNTs and HMWNTs.

Samples	Pseudo-First-Order Models			Pseudo-Second-Order Models				Intraparticle Diffusion Equation		
	k_1_ (min^−1^)	q_e_ (mg g^−1^)	R^2^	k_2_ (g mg^−1^ min^−1^)	q_e_ (mg g^−1^)	h_0_ (mg g^−1^ min^−1^)	R^2^	k_int_ (mg g^−1^ min^−1/2^)	C (mg g^−1^)	R^2^
MWNT-1	0.0111	28.28	0.364	6.54 × 10^−4^	30.30	0.612	0.999	0.5329	11.907	0.945
MWNT-2	0.0076	22.65	0.687	6.54 × 10^−4^	30.29	0.603	0.999	0.6508	8.476	0.881
MWNT-3	0.0092	27.60	0.831	3.19 × 10^−4^	25.64	0.214	0.999	0.4424	7.095	0.944
HMWNT-1	0.1258	18.11	0.210	10.65 × 10^−4^	21.88	0.510	0.999	0.2479	12.478	0.929
HMWNT-2	0.0124	16.50	0.199	3.61 × 10^−4^	20.45	0.151	0.997	0.2256	8.289	0.973
HMWNTs-3	0.0045	17.60	0.886	4.35 × 10^−4^	18.45	0.148	0.996	0.4662	2.203	0.905

**Table 4 nanomaterials-12-01592-t004:** Raman parameters of MWNTs and HMWNTs prior to and after HA adsorption.

Samples	W_G_ (cm^−1^)	W_D_ (cm^−1^)	W_2D_ (cm^−1^)	W_D+G_ (cm^−1^)	I_D_/I_G_	I_2D_/I_D+G_
MWNT-1	1575	1340	2672	2895	1.17	2.24
MWNT-1 + HA	1573	1337	2675	2910	1.20	1.94
MWNT-2	1573	1339	2672	2893	1.01	2.55
MWNT-2 + HA	1576	1341	2675	2895	0.96	2.25
MWNT-3	1568	1340	2673	2922	1.08	2.03
MWNT-3 + HA	1577	1344	2683	2928	1.03	1.72
HMWNT-1	1571	1341	2673	2929	1.10	2.54
HMWNT-1 + HA	1570	1335	2667	2895	1.24	1.72
HMWNT-2	1571	1336	2671	2895	1.07	2.18
HMWNT-2 + HA	1575	1341	2673	2916	0.99	1.97
HMWNT-3	1570	1340	2674	2910	1.12	2.39
HMWNT-3 + HA	1574	1342	2677	2922	1.06	2.08

## Data Availability

Not applicable.

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
