# Peer review of "Effect of Tube Diameters and Functional Groups on Adsorption and Suspension Behaviors of Carbon Nanotubes in Presence of Humic Acid"

_nanomaterials, 2022, doi:10.3390/nano12091592_

Round 1

Reviewer 1 Report

The Authors should account for the following suggestions:

1- The title is too generic and should indicate what sorption behaviour and substance are considered.

2- Transient kinetic analysis should account for interparticle diffusion, pore diffusivities (related to the nanotube diameter and structure), and tortuosity. This point should be better clarified in the introduction and HA adsorption kinetics interpretation.

3- Due to a diffusive control on Adsorption kinetics, data can be plotted as a function of the square root of time, recovering Diffusivity parameters that can be more interestingly correlated to the nanotube morphology and geometrical dimensions. Moreover, this parameter will also depend on the chemical nature of the adsorbing molecules, which is described at equilibrium by the Langmuir and Freundlich adsorption isotherms parameters.

4- The complex equilibrium and adsorption kinetic behaviour is influenced by the Ph  (that acts on the equilibrium concentration), by morphological parameters (such as nanotube multiwall and dimensions), and by the chemical nature of the adsorbing (and diffusing) molecular species. The Authors should discern among these different factors that influence the transient kinetics that are potential tools to understand their specific influence.

All these points should be considered for a more interesting and exhaustive scientific description of the investigated process.

Reviewer 2 Report

Comments from Reviewer

Title: The adsorption and suspension behaviors of carbon nanotubes with different tube diameters and functional groups

The current form's presentation of methods and scientific results is satisfactory for publication in the Nanomaterials journal. The minor and significant drawbacks to be addressed can be specified as follows:

  1. The authors must supply an ORCID ID for all authors. Getting an ORCID iD is FREE, quick and easy to do through the ORCID registration page: https://orcid.org/register. Please, give the respective ORCID ID in the manuscript.
  2. Line 33. Multiwall ---> multiwall – see line 34 (hydroxylated MWNT (HMWNTs)).
  3. (i) Lines 99, 102, and 104. 23-24 ---> 23,24 (ii) Line 102. 29-30 ---> 29,30. (iii) etc. See line 95 correct: [25,26] and line 96 correct: [27,28].
  4. Lines 107 – 116. What about water? What is its impact? In my and others, the interaction between surface groups and water is significant.
  5. Lines 132 and 133. Outer diameter? Inner diameter? The number of walls?
  6. For example, line 173. L-1. L-1 ---> cm-3. See lines 138 and 139: cm… Standardize to the SI system.
  7. Table 1. How was "Carbon distribution" estimated? Why did the authors not use standard methods, such as e.g. XPS or FTIR (or even Boehm's method)?
  8. 1. The images are so small that it's hard to see anything. Do the authors see adsorbed HA for the samples ... + HA?
  9. Line 241, The specific surface areas (BET) of MWNTs and HMWNTs were 129.5 - 342.4 (…). What sample? MWNT-1? MWNT-2? MWNT-3?
  10. Please publish nitrogen adsorption isotherms for test samples in the Supplementary Information. It is essential information for readers.
  11. Line 243, the decrease of tube diameter. How do the authors know about it? TEM pictures? I can't see anything like that!!!
  12. 2. Some points don't keep the trend. How many times was the measurement repeated for a given system?
  13. 2. R?
  14. Line 494. Formula ---> Equation
  15. Table S1. Specific surface area reduction rate? What is this?

Sincerely,

The reviewer.

Reviewer 3 Report

Authors presented adsorption and suspension behavior of CNTs with different functional groups. Some comments to improve the manuscript:

  1. In the experimentation part, how did the authors ascertain the diameter of CNTs? 
  2. In section 3.1, further explanation is required as to why increase in pH decreases Zeta potential.
  3. The authors need to include an appendix explaining clearly the abbreviations used in the manuscript, especially in formulas and Tables.
  4. Major English language corrections are required. It is suggested that proof reading is required with native English language service.
  5. The citations seem to be excessive. It is suggested to cut off unnecessary citations and restrict to limited and relevant literature.

Round 2

Reviewer 1 Report

The Authors revised the paper according to my suggestions

Reviewer 2 Report

My comments have been appropriately addressed in the revised manuscript.

Reviewer 3 Report

Authors have addressed all the queries raised and the manuscript can be accepted for publication.